

# Development and application of 14 microsatellite markers in the burying beetle *Nicrophorus vespilloides* reveals population genetic differentiation at local spatial scales

Sonia Pascoal and Rebecca M. Kilner

Department of Zoology, University of Cambridge, Cambridge, United Kingdom

## ABSTRACT

Burying beetles (genus *Nicrophorus*) are relatively rare among insects in providing sophisticated parental care. Consequently, they have become model species in research analysing social evolution, the evolution of parental care and mating systems. We used the recently published *N. vespilloides* genome and transcriptome to develop microsatellite markers. Specifically, we developed 14 polymorphic markers with five to 13 alleles per locus and used them to investigate levels of genetic differentiation in four south Cambridgeshire (UK) populations of *N. vespilloides*, separated by 21 km at most. The markers revealed significant genetic structuring among populations (global $F_{ST} = 0.023$) with all but one of the pairwise comparisons among populations being significant. The single exception was the comparison between the two closest populations, which are approximately 2.5 km apart. In general, the microsatellite markers showed lower observed heterozygosity than expected. We infer that there is limited dispersal between populations and potentially also some inbreeding within them and suggest that this may be due to habitat fragmentation. We discuss these results in the context of recent laboratory experiments on inbreeding and beetle flight.

## INTRODUCTION

Burying beetles (genus *Nicrophorus*) are relatively unusual among insects in providing elaborate parental care for their developing larvae. Reproduction centres on the fresh carcass of a small vertebrate (like a songbird or mouse), which adults locate by flight. If more than one adult of the same sex finds the carcass, they will commonly fight to determine ownership (e.g., *Otronen, 1988*; *Scott, 1994*). Defeated subordinates may stay nearby: males may sneak matings with the dominant female, while females might become intraspecific brood parasites (e.g., *Müller, Eggert & Dressel, 1990*; *Hopwood et al., 2015*). The dominant individuals of each sex then pair up and together prepare the carcass for reproduction (although mated females can singlehandedly raise young: *Müller et al., 2007*). They remove the fur or feathers, roll the flesh into a ball and bury it below ground in a shallow grave.

Corresponding author
Sonia Pascoal, scm77@cam.ac.uk

During this time, parents defend their carcass breeding resource from attack by conspecifics, congenerics and other carrion-feeding insects (*Robertson, 1993*; *Trumbo, 1990*). Eggs are laid in the soil near the carcass. Newly hatched larvae crawl into a crater on the brood ball and there they solicit attention from their parents, who also stay to protect them from attack (*Trumbo, 2007*). About a week after the larvae hatch, the carcass is entirely consumed. The beetle parents then fly off to search for new mating opportunities or fresh carrion and the larvae disperse into the soil to pupate (*Scott, 1998*).

Burying beetles thus play a key ecological role as nutrient recyclers (e.g., *Royle, Hopwood & Head, 2013*). Furthermore, their relatively unusual natural history, and the ease with which they can be bred in the lab, means that several have become popular study species in experimental analyses of social evolution, the evolution of parental care and mating systems (e.g., *Royle, Hopwood & Head, 2013*; *Scott, 1998*). Nevertheless, despite their widespread use in the lab, relatively little work has focused on the burying beetles' ecology in nature. This is because it is difficult to track marked burying beetles through their life course to understand patterns of dispersal (e.g., *Attisano & Kilner, 2015*), the likely extent of competition for limited carrion resources (e.g., *Kilner et al., 2015*), the degree of connectivity between populations and therefore the potential for inbreeding (e.g., *Pilakouta et al., 2015*). Yet this knowledge is key for interpreting the results of experiments carried out in the laboratory (e.g., *Kilner et al., 2015*; *Attisano & Kilner, 2015*; *Pilakouta et al., 2015*).

Furthermore, habitat fragmentation has recently been suggested to influence population structure in beetles (*Keller & Largiader, 2003*; *Suzuki & Yao, 2014*). Habitat fragmentation by deforestation has specifically been hypothesised to influence dispersal of *N. americanus* in the USA (*Creighton et al., 2009*) and *N. quadripunctatus* populations in Japan (*Suzuki & Yao, 2014*). Yet behavioural work on flight performance shows that burying beetles are capable of sustained flight over tens of kilometres (*Attisano & Kilner, 2015*). Therefore, a key aim of this study was to determine the extent to which burying beetles disperse in nature over relatively short distances, using genetic techniques (cf *Houston et al., 2015*).

Recently, molecular resources have been developed for *N. vespilloides*, including a genome, epigenome (*Cunningham et al., 2015*) and transcriptomes (e.g., *Parker et al., 2015*; *Palmer et al., 2016*). We took advantage of these newly available molecular tools to develop microsatellite markers. We used the markers to determine the extent of genetic differentiation between natural populations of *N. vespilloides*, deliberately choosing sites on a local scale, no more than 21 km apart (Fig. 1) to test the power of the markers we developed. These four populations were all in south Cambridgeshire and at: Waresley Woods (Latitude: 52.17487°; Longitude: −0.17354°), Gamlingay Woods (Latitude: 52.15555°; Longitude: −0.19286°), Byron's Pool (Latitude: 52.17305°; Longitude: 0.10196°) and Madingley Woods (Latitude: 52.22658°; Longitude: 0.04303°). These four populations inhabit patches of woodland that are islands in a landscape dominated by arable farmland and urban development. Their close proximity enabled us to determine whether habitat fragmentation or beetle flight performance could better explain population structure on a local scale.

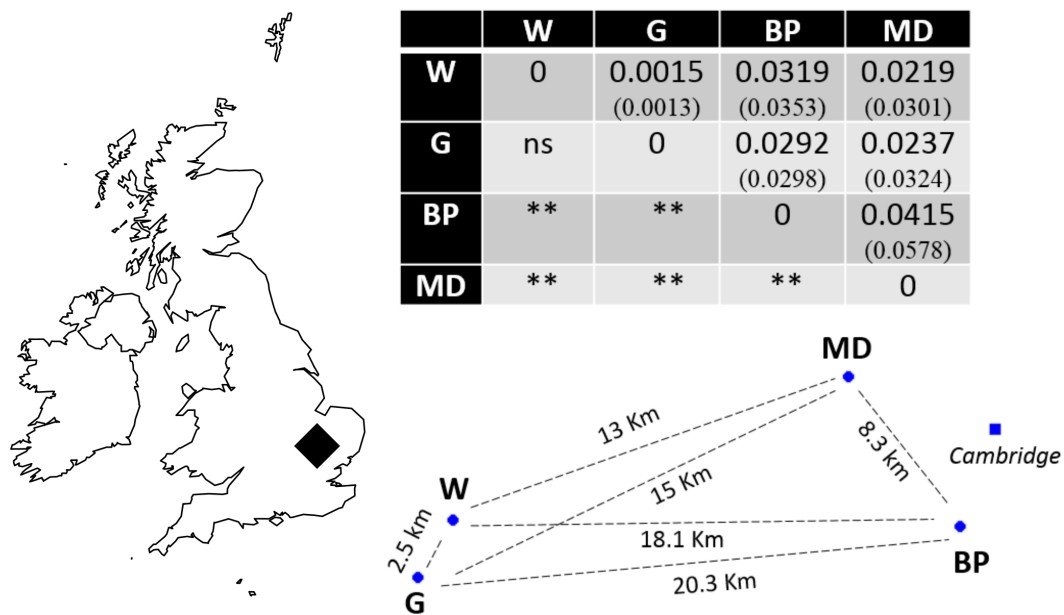

**Figure 1** **Sampling location in south Cambridgeshire and genetic differentiation in pairwise comparisons between populations.** $P$ values for pairwise $F_{ST}$ estimates across all loci are indicated below the diagonal, after Bonferroni correction ($\alpha = 0.008333$). Values in brackets represent the pairwise $F_{ST}$ values using only the 9 markers in HWE; the significance levels were identical in both runs. W, Waresley; G, Gamlingay; BP, Byron's Pool; MD, Madingley. Distances are "as the crow flies" distances calculated using online tools. The map and population's spatial representation (using the sites geographical coordinates) were produced in R.

## MATERIAL AND METHODS

### Beetles

Adult beetles from the four south Cambridgeshire populations (Fig. 1): Gamlingay ($n = 40$), Waresley ($n = 40$), Byron's Pool ($n = 33$) and Madingley ($n = 26$) were collected during May–October 2016 using Japanese beetle traps. Male and female beetles, sampled in equal numbers, were brought back to the laboratory alive and preserved in absolute ethanol for genetic analysis. In order to avoid collecting related individuals, periodical sampling was performed and only adult beetles (not larvae) were analysed. These were males and females attracted at random to the mouse carcasses provided in the beetle traps.

In an earlier pilot study, we tested microsatellite markers from *N. quadripunctatus* (*Suzuki & Yao, 2014*), in *N. vespilloides*, but these failed to amplify reliably (M Schrader, pers. comm., 2015). We therefore developed new markers specifically for *N. vespilloides*.

### DNA extraction, microsatellite amplification and analysis

Total genomic DNA ($n = 139$) was extracted individually from beetle heads using the DNeasy Tissue Kit (Qiagen, Hilden, Germany). Molecular markers were developed with the program msatcommander (*Faircloth, 2008*), using the publicly available *N. vespilloides* genome (NCBI Bioproject number PRJNA284849; *Cunningham et al., 2015*) and transcriptome (NCBI Bioproject PRJNA285436; *Parker et al., 2015*). Twenty potential microsatellites were tested and optimised. From these, 14 polymorphic microsatellite

markers were amplified using three primer mixes with the Qiagen Multiplex PCR kit, following the manufacturer's instructions, to a final volume of 10 µL. The fluorescent M13 tail single-reaction nested PCR method (*Schuelke, 2000*) using four tails (*Tysklind, 2009*) was used to amplify the loci. An initial denaturation step of 15 min at 95 °C was followed by 13 cycles at 94 °C for 30 s, 62 °C for 90 s, and 72 °C for 60 s. In order to attach the dye tails to the PCR product, an extra 31 cycles at 94 °C of 30 s, 50 °C for 90 s and 72 °C for 60 s were performed and followed by a final extension at 60 °C for 30 min. Extension products were resolved on an ABI 3730 instrument at the Edinburgh Genomics Institute Sanger Sequencing Centre with GeneScan 500 LIZ (Applied Biosystems, Foster City, CA, USA) as internal size standard. Alleles were scored and checked using Peak Scanner v.1.0 (Applied Biosystems).

GenAlEx version 6.5 (*Peakall & Smouse, 2012*) and FSTAT (*Goudet, 1995*) were used to generate descriptive statistics (e.g., number of alleles, allelic frequencies, mean number of alleles per locus and observed ($H_0$) and expected heterozygosity ($H_E$)). Tests for deviations from Hardy-Weinberg proportions and genotypic linkage equilibrium were estimated using GENEPOP (*Raymond & Rousset, 1995*; *Rousset, 2008*). CERVUS v3.0.7 (*Kalinowski, Taper & Marshall, 2007*) was used to test for null alleles. Estimates of $F_{ST}$, $F_{IS}$, population pairwise $F_{ST}$ and their significance per population over all loci were calculated using FSTAT (*Goudet, 1995*).

## RESULTS

### Microsatellite development

We screened the *N. vespilloides* genome (4,660 contigs; NCBI Bioproject number PRJNA284849) for microsatellite markers with at least 8 repeats for di- and trinucleotides and at least 6 repeats for tetra-, penta- and hexanucleotides, and identified 1818 sequences containing repeats. A total of 5,547 microsatellites were present and 4515 primer pairs were obtained using Primer3 (*Rozen & Skaletsky, 2000*) incorporated into msatcommander (*Faircloth, 2008*). Similar searches to the transcriptome identified 263 microsatellites and 69 primer pairs were designed. To maximize the potential for amplification, we rejected primers of low quality, and that were likely to self-anneal. To facilitate multiplexing, we chose markers from 100–500 bp that varied in size and the number of repeat motifs. To avoid linkage, we chose sequences with just one marker. Twenty microsatellites were chosen for molecular marker optimization (16 from the genome and four from the transcriptome). From these, a robust suite of 14 reliable microsatellites was derived for the population genetic analysis (see Supplementary Information). By 'robust' we mean that they (i) amplified reliably in all populations and in the majority of the samples; (ii) did not show secondary amplification; (iii) were polymorphic in all populations; and (iv) were relatively easy to score (e.g., did not show stuttering).

### Microsatellite analysis

All of the 139 *N. vespilloides* collected from the four Cambridgeshire populations were genotyped for the 14 microsatellites. All 14 loci were polymorphic for the tested populations and the number of alleles per locus ranged between five and 13, with a total number of 134

alleles in the global sample (Table 1). The level of genetic variability was similar across loci. The expected heterozygosity ($H_E$) per locus ranged from 0.272 to 0.841, and the observed heterozygosity ($H_O$) ranged from 0.247 to 0.825 (Table 1). All individual loci but one showed lower observed heterozygosity than expected heterozygosity (over all loci $H_E = 0.714$ and $H_O = 0.652$). The expected heterozygosity across all loci per population ranged from 0.696 to 0.706, while the observed heterozygosity ranged from 0.638 to 0.679 (Table 2).

Tests for concordance with Hardy-Weinberg equilibrium (HWE) revealed deviations from HWE in locus Nvesp_D, Nvesp_F, Nvesp_I, Nvesp_G and Nvesp_H (Table 1). Testing HWE for individual populations and loci revealed that this disequilibrium remained significant within populations (Nvesp_D: significant in all populations but Waresley; Nvesp_F: significant for Byron's Pool only; Nvesp_I: significant in Waresley only; Nvesp_G: significant in Waresley only; Nvesp_H: significant in Gamlingay and Byron's Pool) (Table 2). The frequency of null alleles was low across loci (Table 1) and, overall, close to zero (indicating absence of null alleles). However, three of the loci exhibiting deviations from HWE (Nvesp_D, Nvesp_I and Nvesp_H) showed some of the highest null allele frequencies (>0.05).

Evidence of linkage disequilibrium was observed in pairwise loci Nvesp_Q/Nvesp_E and Nvesp_E/ Nvesp_G in the global population test. Global $F_{IS}$ value was 0.085, suggesting some heterozygous deficiency. A pattern of genetic differentiation (global $F_{ST} = 0.023$) was observed in the sampling area, with all but one significant population-pairwise $F_{ST}$ after Bonferroni correction ($\alpha = 0.008333$). In order to assess whether there were potential biases in the markers that exhibit deviations from HWE or the presence of potential null alleles, the analyses were run with and without these markers. We found that the two runs rendered similar results (Fig. 1).

## DISCUSSION

We developed microsatellite markers to infer details of the burying beetle's ecology that cannot be deduced through simple observation, but which are becoming increasingly important for the interpretation of experiments on this species in the laboratory. The process of microsatellite development was greatly facilitated by the existing *N. vespilloides* genome (*Cunningham et al., 2015*) and transcriptome (*Parker et al., 2015*). The available genomic tools, however, are still poorly annotated and so further detailed characterisation of the sequences containing the markers is still limited (see Supplementary Information). Nevertheless, our analyses suggest that the markers are predominantly unlinked. We rapidly developed a set of 14 (out of 20) reliable polymorphic markers for the species: i.e., 70% were successful. This proportion of successful markers is similar to that obtained for *N. quadripunctatus* (*Suzuki & Yao, 2014*), although these authors used enriched genomic libraries for marker development.

Our genetic analyses revealed significant deviations from Hardy-Weinberg equilibrium at five loci. This is most likely due to an excess of homozygotes at these loci but could also be due to the presence of null alleles in three of these markers. The only other marker with a putatively high null allele frequency (>0.05) was Nvesp_E. However, the high

**Table 1** **Main genetic variability measures by locus of *N. vespilloides* from Cambridgeshire.** $T$ (°C), annealing temperature; bp, base pairs; G, genome; T, transcriptome; Na, number of alleles found per locus; $H_E$, expected heterozygosity; $H_O$, observed heterozygosity; $F_{IS}$, standardized genetic variance within populations at each locus; $F_{ST}$, standardized genetic variance among populations at each locus; Null, frequency of null alleles per locus; HW, Hardy-Weinberg $P$ values.

| Locus | Primer sequence 5′–3′ | Product size (bp) | Repeat motif | $T$ (°C) | PCR | Source | Na | $H_E$ | $H_O$ | $F_{ST}$ | $F_{IS}$ | Null | HW |
|---|---|---|---|---|---|---|---|---|---|---|---|---|---|
| Nvesp_A | F: Fam-CTACGGCGTGCAGAATTACC R: AACTCTCTGGTGTCGACGTC | 138 | (AAC)9 | 62 | Mix1 | G | 13 | 0.757 | 0.770 | 0.002 | −0.026 | −0.0114 | 0.459 |
| Nvesp_D | F: Pet-TACGTGCGGTAATGAGGCG R: ACGCCCTGCTCCCTATTTAG | 201 | (AAC)11 | 62 | Mix1 | G | 8 | 0.829 | 0.585 | 0.016 | 0.287 | 0.1701 | ***P* < 0.001** |
| Nvesp_J | F: Vic-TGTGTGTAGAGTGGACGGG R: TGGACGAGTTGAAGACGAGG | 303 | (AAAG)7 | 62 | Mix1 | G | 9 | 0.657 | 0.631 | 0.035 | 0.010 | 0.0131 | 0.554 |
| Nvesp_M | F: Ned-CCAGCAACCCACAAAGAAGC R: ATACCACAAGTCCCGACCTG | 373 | (AG)10 | 62 | Mix1 | G | 11 | 0.841 | 0.825 | 0.019 | 0.039 | 0.0244 | 0.058 |
| Nvesp_Q | F: Fam-ATGCGGCTTTGATATCCAGG R: TCAGATTCCGCTCTCCTTCC | 428 | (AAT)8 | 62 | Mix1 | G | 8 | 0.516 | 0.494 | 0.005 | 0.075 | 0.0560 | 0.140 |
| Nvesp_B | F: Fam-GTTGTTTCCGGTTGTTTGCG R: TTCGAAGTTAAACGGCCGTG | 158 | (AC)8 | 62 | Mix2 | G | 10 | 0.723 | 0.701 | 0.021 | 0.035 | 0.0266 | 0.496 |
| Nvesp_F | F: Pet-TAAAGGGTTGGGAGGTTGGC R: CACGATCCATACACGTGCAC | 216 | (AC)10 | 62 | Mix2 | G | 10 | 0.802 | 0.723 | 0.023 | 0.075 | 0.0458 | **0.004** |
| Nvesp_I | F: Vic-CTGATCACCGGAACCCTCTC R: GAATTCCCGGGTTTATGCCG | 286 | (AG)8 | 62 | Mix2 | T | 6 | 0.569 | 0.498 | 0.006 | 0.135 | 0.0793 | **0.036** |
| Nvesp_P | F: Fam-TGGTGATGCAATTGTGAGGC R: CGGTTGGCAGACGATGTAAC | 410 | (ATC)8 | 62 | Mix2 | G | 9 | 0.809 | 0.741 | 0.028 | 0.082 | 0.0498 | 0.189 |
| Nvesp_E | F: Pet-ATGGATGGATGGAGAGTGGC R: TTGATGGTTTCGAAAGGGCG | 201 | (AC)8 | 60 | Mix3 | T | 11 | 0.787 | 0.680 | 0.092 | 0.139 | 0.1109 | 0.182 |
| Nvesp_G | F: Fam-CGTGTGCGTGTTTCTACCTC R: ATGGGCACGTATCCATACCC | 224 | (AT)8 | 60 | Mix3 | T | 12 | 0.834 | 0.776 | 0.013 | 0.064 | 0.0351 | **0.009** |
| Nvesp_H | F: Vic-TCGTAGATGTCTCGTGCCTG R: CAGTTTGAAGGTGGTGGCTG | 283 | (AG)9 | 60 | Mix3 | G | 12 | 0.840 | 0.737 | 0.013 | 0.120 | 0.0669 | ***P* < 0.001** |
| Nvesp_K | F: Ned-GCTCTCATTCTCCCAAACGC GTGGACGCGCATAAGTTGTC | 334 | (AGG)8 | 60 | Mix3 | G | 5 | 0.272 | 0.247 | −0.002 | 0.085 | 0.0328 | 0.260 |
| Nvesp_O | F: Fam-ATGCCAATTAACGCGTCGAG CATCGTTACCTGTGCGACTG | 395 | (AAG)8 | 60 | Mix3 | G | 10 | 0.760 | 0.718 | 0.013 | 0.054 | 0.0387 | 0.085 |
| All | | | | | | | 134 | 0.714 | 0.652 | 0.023 | 0.085 | | |

Pascoal and Kilner (2017), *PeerJ*, DOI 10.7717/peerj.3278

**Table 2  Main genetic variability measures for Cambridgeshire populations.** W, Waresley; G, Gamlingay; BP, Byron's Pool; MD, Madingley; $N$, mean number of samples per locus; $N$-all, mean number of alleles per locus; $H_E$, expected heterozygosity; $H_O$, observed heterozygosity; A–O refers to Nvesp_A-Nvesp_O.

| Pop | $N$ (±SD) | $N$-all (±SD) | $H_E$ (±SD) | $H_O$ (±SD) | Hardy-Weinberg $P$ values | | | | | | | | | | | | | |
| --- | --- | --- | --- | --- | A | D | J | M | Q | B | F | I | P | E | G | H | K | O |
| W | 38.143 (0.619) | 7.571 (0.453) | 0.699 (0.047) | 0.679 (0.046) | 0.255 | 0.141 | 0.632 | 0.228 | 0.529 | 0.180 | 1.000 | **0.009** | 0.834 | 0.328 | **0.004** | 0.297 | 1.000 | 0.544 |
| G | 37.214 (0.639) | 7.714 (0.474) | 0.706 (0.043) | 0.638 (0.045) | 0.665 | **0.000** | 0.969 | 0.115 | **0.036** | 0.267 | 0.535 | 0.282 | 0.164 | 0.221 | 0.351 | **0.000** | 0.064 | **0.028** |
| BP | 29.714 (0.485) | 7.429 (0.581) | 0.702 (0.043) | 0.664 (0.049) | 0.428 | **0.016** | 0.908 | **0.022** | 0.191 | 0.855 | **0.000** | 0.206 | 0.256 | 0.083 | 0.136 | **0.017** | 0.436 | 0.904 |
| MD | 19.286 (0.606) | 6.429 (0.343) | 0.696 (0.044) | 0.649 (0.045) | 0.288 | **0.037** | 0.059 | 0.936 | 0.598 | 0.607 | 0.102 | 0.518 | 0.103 | 0.571 | 0.183 | 0.100 | 0.232 | 0.071 |

number of homozygotes for this marker is unlikely to be due to null alleles because it is in HWE (*Kalinowski, Taper & Marshall, 2007*). We are confident that our results are not biased by including markers that deviated from HWE, or potential null alleles, because we obtained similar results when we excluded these markers from our analyses. In general, the microsatellite markers showed lower observed heterozygosity than expected and the global $F_{IS}$ value (0.085) also suggested some heterozygosity deficiency. We infer from these findings that there is limited gene flow between our study populations, and potentially some inbreeding as well. The consequences of inbreeding in *N. vespilloides* have recently been analysed experimentally in the laboratory (e.g., *Mattey, Strutt & Smiseth, 2013*; *Pilakouta et al., 2015*; *Pilakouta & Smiseth, 2016*) but to our knowledge, our study provides the first indication that *N. vespilloides* might breed with relatives in nature and it matches previous results obtained for *N. quadripunctatus* (*Suzuki & Yao, 2014*).

Consistent with our interpretation of limited gene flow between populations, we found significant pairwise population differentiation between all but one pair of populations (Waresley-Gamlingay; approximately 2.5 km apart) despite the low geographical separation between the sampling sites (maximum 20.3 km). This genetic differentiation may be the result of neutral population differentiation, or the effects of selection acting on functional genes correlated with the neutral markers (e.g., *Rousset & Raymond, 1995*). We think the former possibility is more likely because we obtained similar results when we ran the analyses with and without markers in HWE and because a BLASTx search showed that the markers did not generally correspond to (albeit limited) known coding genes in *N. vespilloides.*

Previous work on flight performance of *N. vespilloides* showed that these beetles have a wide distribution of flight distances in the laboratory ranging from 68 m to 26 km. Furthermore, flight durations ranged from 61 s to 6.5 h under laboratory conditions (*Attisano & Kilner, 2015*). Beetles tethered in flight mills may be able to fly for greater distances than naturally flying beetles because they bear less of their weight in flight. But even if we assume that beetles in natural flight cover only a third of the distance that they achieved in a flight mill, our data suggest that population differentiation within the sampling area is not solely attributable to the flight range of the burying beetle.

We suggest instead that habitat fragmentation has driven the fine-scale population structure we report here by imposing barriers that limit dispersal (cf *Pascoal et al., 2009*; *Kanno, Vokoun & Letcher, 2011*; *Valtonen et al., 2014*). Our analyses suggest that each 'island' population of burying beetles is increasingly reproductively isolated with increasing geographic distance between woodlands. Perhaps *N. vespilloides* is unwilling, or unable, to undertake flights across open fields and through housing. Studies of congeneric species (*Nicrophorus marginatus*, *N. tomentosus*, *N. orbicollis* and *N. defodiens*) found that the size of these woodland fragments affects the abundance and reproductive success of the resident burying beetle population (*Trumbo & Bloch, 2000*; *Gibbs & Stanton, 2001*). Our analyses suggest that the extent of their connectivity might also be an important factor for promoting gene flow and preventing populations from becoming smaller and more inbred. If our interpretation is correct, then it has important conservation implications because it suggests that there is a threshold size of woodland required to sustain an outbreeding burying beetle population. It also might help explain why populations of American burying

beetles (*N. americanus*) have collapsed so spectacularly in recent years following substantial deforestation, causing it to become endangered (*Anderson, 1982*; *Creighton et al., 2009*).

Although Single Nucleotide Polymorphism (SNPs) are starting to be the tool of choice for population genetics/genomics studies, microsatellites still provide a cheaper alternative. Nevertheless, we anticipate that the microsatellites we have developed will prove most useful in future work for assigning parentage (S Pascoal, 2016, unpublished data) because the large brood size typically seen in *N. vespilloides* still makes other techniques prohibitively expensive. *N. vespilloides* mates rampantly and promiscuously in the laboratory (e.g., *House et al., 2009*) and in natural populations (e.g., *Müller et al., 2007*). Although several previous studies have analysed strategies used by males for securing paternity (e.g., *Eggert, 1992*; *House et al., 2009*), parentage has never before been assigned using microsatellites. The microsatellites we have developed here thus pave the way for more detailed analyses of the evolutionary causes and consequences of promiscuity in this species.

## ACKNOWLEDGEMENTS

We thank Benjamin Jarrett, Syuan-Jyun Sun and Sue Aspinall for providing the field-caught beetles for the genetic analysis.

### Funding
This project was supported by a Consolidator's Grant from the European Research Council (310785 Baldwinian_Beetles) and a Royal Society Wolfson Merit Award, both to RMK. The funders had no role in study design, data collection and analysis, decision to publish, or preparation of the manuscript.

### Grant Disclosures
The following grant information was disclosed by the authors:
European Research Council: 310785 Baldwinian _Beetles.
Royal Society Wolfson Merit Award.

### Competing Interests
The authors declare there are no competing interests.

### Author Contributions
- Sonia Pascoal conceived and designed the experiments, performed the experiments, analyzed the data, wrote the paper, prepared figures and/or tables.
- Rebecca M. Kilner conceived and designed the experiments, contributed reagents/materials/analysis tools, wrote the paper.

### Data Availability
The raw data has been supplied as Data S1.

## Supplemental Information

Supplemental information for this article can be found online at http://dx.doi.org/10.7717/peerj.3278#supplemental-information.

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
