# Peer review of "Development and application of 14 microsatellite markers in the burying beetle Nicrophorus vespilloides reveals population genetic differentiation at local spatial scales"

_PeerJ, doi:10.7717/peerj.3278_

## Round 0.1 · original submission · Major Revisions

Two reviewers have assessed this manuscript, and their recommendations were quite different. However, they both identified a number of major issues that would need to be addressed.

This paper as it stands is little more than a primer note, with very minimal analysis of the data.

I recommend that the authors take on the suggestions of both reviewers and produce a more robust paper in the process.

Reviewer 1 ·

Basic reporting

This article describes the development of 14 microsatellite markers for the burrowing beetle N. vespilloides, and uses a fine scale case study to demonstrate their use for identifying genetic structure between populations. The results presented here represent a primer note and are somewhat outside the scope of the journal. It may be more appropriate in PeerJ Preprints or a relevant Resources journal.

The article is written in clear English and is of a generally high standard.
Specific comments on writing/citations - sentences on lines 53, and 68-71 could be deleted.
Some instances in introduction where statements required citations (lines 40 and 43 for example).
Lines 175-184 - This explanation is quite vague and difficult to follow - needs to be clarified

Experimental design

If the article is to be considered as an original research article addressing a question about fine-scale population structure then the question and background information surrounding it need to be further developed. Interesting questions have been raised in the discussion as to how habitat fragmentation may influence population structure (lines 164-168) and the system appears to lend itself to addressing this.

More information needs to be included in methods as to how samples were collected. Were random samples of populations taken? Were steps taken to reduce chance of sampling siblings in the field? If it is suggested that these populations are suffering from inbreeding then sampling individuals from the same nests would influence the results commented on in lines 150-152.

Validity of the findings

Deviations from Hardy-Weinberg Equilibrium have been identified in 5 out of 14 microsatellite loci, however it is unclear whether these loci were included in analyses of population structure that followed. These analyses should be done with and without loci that exhibit deviations from HWE in order to show whether these loci are driving the observed patterns. Also, tests for null alleles are not presented, and their presence may be causing the consistently higher than expected rates of observed homozygosity seen here.
Also, as per line 158, if beetles can travel up to 26km, yet all population pairs are within this range then finding significant differences in Fst is not an expected result, and could be potentially the result of inclusion of loci that deviate from HWE or have null alleles present.

Reviewer 2 ·

Basic reporting

Very well written, suffiecnet background and self contained article. The only issue is what constitutes raw data for microsatellites, espcially microsatellite development. I suggest the authors should supply more detail about the origins of the markers (simply G or T is probably not enough information), and I would suggest that some indcation of the chromatograms is also require. Despite the authors' assertions that microsatellites are straightforward - there are definite issues that genome sequencing can help to resolve.

Experimental design

The rsearch is within the aims and scope, the question is well defined and investigated. However there are a couple of gaps that could be addressed.
Firstly, as stated above the markers themselve need to be better described - not all studies have access to genomes and transcriptomes so I would expect a much more rigorous explanation of the markers.
Secondly there is not enough discussion of why the 20 initial markers were chosen and why "robust" suite of 14 markers is robust - - what does a robust marker look like?

Validity of the findings

The findings are limited although valid.
The authors have however neglected to investigate the neutrality of their markers and whether selection might be driving the lack of HWE - again with by using the genomic resources at hand to give a better description of the markers (location? coding? linkage?), then they can help to remove the speculation around inbreeding.
The authors also speculate about the use of these markers in parentage studies -they have gifted themselves with a fantastic study organism (genomic resources and simplicity of lab breeding) so why not actually test their suitability for parantage analyses - it would not require a huge additional amount of genotyping.
Finally, paragraph 175-184 in the discussion is pure speculation - it is impossible to make any findings on female sstrategically reallocating resources from this small data set.

Additional comments

Overall a neat exploration of the microsats, with some indication that there is an intersting story amongt the populatiopns - so its a worthwhile contribution. I just think you've missed the boat here a little with the genomic resources at your disposal to give a much better description of your chosen markers - which will help to make a better choice as to the validity of them as future markers and the findings of this preliminary study.

---

## Round 0.2 · accepted · Accept

This paper has been substantially revised and the reviewer comments have been satisfactorily addressed.